# Application and Evaluation of the Antifungal Activities of Glandular Trichome Secretions from Air/Sun-Cured Tobacco Germplasms against *Botrytis cinerea*

**DOI:** 10.3390/plants13141997

**Published:** 2024-07-22

**Authors:** Jing Liu, Jiao Wang, Yongmei Du, Ning Yan, Xiao Han, Jianhui Zhang, Yuqing Dou, Yanhua Liu

**Affiliations:** 1Plant Functional Ingredient Research Center, Tobacco Research Institute of Chinese Academy of Agricultural Sciences, 11 Keyuanjingsi Road, Laoshan, Qingdao 266101, China; liujing05302021@hotmail.com (J.L.); duyongmei@caas.cn (Y.D.); yanning@caas.cn (N.Y.); 46326099@163.com (X.H.); 2School of Life Sciences, Ludong University, Yantai 264025, China; wangj362@nenu.edu.cn; 3Tobacco Science Institute of Sichuan Province, Chengdu 610094, China; sgg94@126.com

**Keywords:** air/sun-cured tobacco germplasms, glandular trichome secretions, quantitative and qualitative analysis, antifungal activity, OPLS analysis

## Abstract

The secretions of the glandular trichomes of tobacco leaves and flowers contain abundant secondary metabolites of different compounds, such as cebradanes, labdanes, and saccharide esters. These secondary metabolites have shown interesting biological properties, such as antimicrobial, insecticidal, and antioxidant activity. In this study, 81 air/sun-cured tobacco germplasms were used as experimental materials. Quantitative and qualitative analyses of the glandular secretion components were conducted using ultra-performance liquid chromatography–quadrupole-time of flight-mass spectrometry (UPLC-Q-TOF MS) and gas chromatography–mass spectrometry (GC-MS). The ethanol extracts of glandular trichomes from tobacco leaves and flowers were evaluated for antifungal activity against the fungus *Botrytis cinerea* using the mycelial growth rate method. Orthogonal Partial Least Squares (OPLS) analysis was then performed to determine the relationship between the trichome secretion components and their anti-fungal activity. The results showed significant differences among the antifungal activities of the tested ethanol extracts of tobacco glandular trichomes. The inhibition rates of the upper leaves and flower extracts against *B. cinerea* were significantly higher than those of the middle and lower leaves, and 59 germplasms (73.75% of the tested resources) showed antifungal rates higher than 50%, with four germplasms achieving a 95% antifungal rate at the same fresh weight concentration (10 mg/mL). The OPLS analysis revealed that the antifungal activity was primarily associated with alpha-cembratriene-diol (α-CBT-diol (Peak7)) and beta-cembratriene-diol (β-CBT-diol (Peak8)), followed by sucrose esters III (SE(III)) and cembratriene-diol oxide. These findings help identify excellent tobacco germplasms for the development and utilization of botanical pesticides against fungi and provide a theoretical reference for the multipurpose utilization of tobacco germplasms.

## 1. Introduction

*Botrytis cinerea* is a fungal pathogen that prefers low temperatures and high humidity, and it is mostly associated with temperate and subtropical regions [1,2]. The pathogen affects more than 200 crops and significantly affects their yield and quality [3]. Recent technological advancements in agricultural planting level and production intensity have greatly accelerated disease transmission [4], resulting in global economic losses of hundreds of billions of dollars each year [5]. *B. cinerea* is a typical gray mold disease with extensive genetic variation, strong pathogenicity, and differentiation ability [1,2]. Currently, more than 20 species of *B. cinerea* have been identified, and nearly 300 crops are at risk of infection [3,6,7,8,9,10]. Due to its characteristics, *B. cinerea* has been classified as ‘highly resistant’ by the Fungicide Resistance Action Committee [11].

At present, disease resistance breeding, ecological regulations, biological control, and chemical agents are used to prevent and control gray mold. In particular, chemical pesticides have been widely used because they are affordable, quick, and easy to use. However, the overuse of chemical pesticides causes the ‘3Rs’ (residue, resistance, and rampant) and has serious impacts on the ecological environment, biodiversity, and human and animal health [12]. Due to this negative impact, plant-derived secondary metabolites represent a safer alternative because they are characterized by low toxicity, residues, pollution, and easy degradability; thus, they have gradually become a trending research topic worldwide. Plant-derived antifungal agents are natural compounds with antifungal activity obtained via physical or chemical reagent extraction, and their application inhibits the growth and reproduction of pathogens while enhancing the natural preservation of food [13,14,15,16,17,18]. Exploiting alternatives to chemical pesticides would be of great value to industrial development [17,19,20].

Tobacco (Nicotiana; Tobacum L. (Solanaceae)) is one of the most widely cultivated plants in the world and has high biomass yield in nature (up to 100 t of leaf biomass per hectare) [21]. The whole plant has dense glandular trichomes with abundant secondary metabolite secretions, including cebradanes, labdanes, and glycolipid compounds [22]. The secretions (0.5–10% fresh weight) present biological activity, including antifungal [23], insecticidal, and antitumor activity [24,25]. Cebradanes are a class of macrocyclic diterpenes that are the most abundant secondary metabolites in the glandular trichome secretions of tobacco germplasms. Studies have shown that cebradanes can inhibit the spore germination of fungi, such as tobacco downy mildew and gray mold, and constrain the growth of bacteria, such as Bacillus subtilis, Staphylococcus aureus, and Proteus vulgaris [26,27]. They also show tobacco antivirus activity (TMV) and anti-insect activity [28,29]. Root defenses in tobacco, tomato, and Arabidopsis thaliana can be induced by the exogenous application of waf1, which inhibits the incidence of Fusarium wilt. The endogenous protein signal of waf1 in tobacco leaves can also be increased to enhance the defense and resistance to TMV [30]. In addition, tobacco sucrose esters also show excellent anti-insect activity against aphids and sweet potato whiteflies [31], and they are environmentally friendly and have high biological safety. The number of conserved tobacco germplasm resources in China ranks first in the world [32], and abundant and high-yielding metabolites have been identified that have important application value in developing natural bacteriostatic agents and reducing botanical pesticide costs. In this study, the effects of tobacco glandular trichome extracts from 81 air/sun-cured tobacco samples on *B. cinerea* were systematically evaluated. Orthogonal Partial Least Squares (OPLS) association analysis was performed to elucidate the inhibitory effects and contributions of different compounds in the extracts of *B. cinerea*. The selected tobacco germplasms can provide an important material basis for the development and utilization of natural fungicides against *B. cinerea*.

## 2. Results

To expand the development and utilization of tobacco germplasm resources in the prevention and control of fungal diseases via plant pesticides, this study systematically evaluated the composition, content, and antifungal activities of ethanolic extracts from 81 tobacco glandular trichomes against *B. cinerea*. The content and antifungal activity of glandular trichome secretions from different tobacco parts and germplasms were significantly different, with the highest content and antifungal activity against *B. cinerea* observed in the flower glandular trichome extracts and the extracted content of different air/sun-cured tobacco germplasm varying from several to hundreds of times. Furthermore, unlike flue-cured tobacco germplasm, the air/sun-cured tobacco was rich in cis-abiethanol, SE(III), and SE(IV). A total of 16 compounds were identified from the glandular trichome secretions of air/sun-cured tobacco germplasm. Among them, compound **T4** has not been previously reported in tobacco.

The antifungal rates of the glandular trichome extracts from 80 different tobacco germplasms varied widely (13.93–100.00%). Four air/sun-cured tobacco germplasms showed inhibition rates above 95%, and they included two local and two introduced germplasms. In addition, the inhibition rates of the local germplasms Huangmaoyan (X79) and Baimaoyan (X80) were 99.42% and 100%, respectively, which were significantly higher than those of the two introduced germplasms Xuejia5 (X63) and Ha20 (X72). Association analysis showed that α- and β-CBT-diols had significant antifungal activities and the largest positive contributions, followed by SE(III) and **T9**, while **T6** had the smallest positive contribution. The results can provide a material basis and technical support for the development of plant bacteriostatic agents and the efficient utilization of sun/air-cured tobacco germplasm.

### 2.1. Assays of Tobacco Glandular Trichome Extract Fractions

#### 2.1.1. UPLC-Q-TOF MS Assay

Configurations of the air/sun-cured tobacco germplasm resources (tobacco glandular trichome exudates) were qualitatively and quantitatively analyzed. The qualitative analysis was performed with UPLC-Q-TOF-MS in the ESI-positive mode The four standards α-cembratrienediol (α-CBT-diol), β-cembratrienediol (β-CBT-diol), cembratrienol (CBT-ol), and cis-abiestrol are shown in Figure 1a. The results showed that 11 common peaks (Peak1–Peak11) occurred in the UPLC chromatograms within 5–8 min (Figure 1b). Compared with the standards, Peak7 corresponds to α-CBT-diol, Peak8 corresponds to β-CBT-diol, Peak10 corresponds to cis-abiestrol, and Peak11 corresponds to CBT-ol. Regarding the quantitative analysis, the contents of the components were calculated by using corresponding standard curves. The contents of the other components without standards were calculated by the semi-quantitative method based on the above formula. The contents of the identified components of the 80 air/sun-cured tobacco germplasms are presented in the attached Table A1.

The retention times, formulae, mass errors, and fragment ions for each component are presented in Table 1. After analyzing the MS data, the molecular formulae were calculated using the exact mass and unsaturation (macrocycle and double bonds) degree of each compound. In addition, the number and type of oxygen-containing substituents in the compounds were calculated according to the fragments. According to the UPLC-Q-TOF-MS chromatograms and fragment ions of the standards α-cembratrienediol, β-cembratrienediol, and cis-abiestrol, compounds **T7** (*m*/*z* 329.2453, [M+Na]^+^), **T8** (*m*/*z* 329.2451, [M+Na]^+^), **T10** (*m*/*z* 313.2446, [M+Na]^+^), and **T11** were identified as α-CBT-diol, β-CBT-diol, cis-abienol, and CBT-ol, respectively.

Compound **T1** (*m*/*z* 361.2341, calculated for C_20_H_34_O_4_Na) showed fragments at *m*/*z* 303.2304 [M-2H_2_O+H]^+^ and *m*/*z* 287.2356 [M-HOOH-H_2_O+H]^+^ and was identified as hydroper-oxy cembratriene-diol. **T2** and **T3** were epimers of **T1** according to the fragment ion *m*/*z* values. Compound **T4** had a sodium adduct ion at *m*/*z* 407.2768 (calculated for C_22_H_40_O_5_Na), dehydrated molecules at *m*/*z* 349.2734 [M-2H_2_O+H]^+^ and *m*/*z* 331.2226 [M-3H_2_O+H]^+^, and a decarboxylated molecule at *m*/*z* 286.2242 [M-3H_2_O-COOH+H]^+^. This indicated that three hydroxyls and one carboxyl were substituted in **T4**. Thus, **T4** was speculated to be seco-cembra trihydroxy-diene acid. Compound **T5** showed a sodium adduct ion at *m*/*z* 345.2492 (calculated for C_20_H_34_O_3_Na) and dehydrated molecules at *m*/*z* 305.2476 [M-H_2_O+H]^+^, *m*/*z* 287.2361 [M-2H_2_O+H]^+^, and *m*/*z* 269.2351 [M-3H_2_O+H]^+^. The fragmentation behavior was similar to the standards of the CBT-diols; therefore, **T5** was presumed to be cembratriene-triol according to the four unsaturations and three hydroxyls. Similarly, the substituent of **T9** (*m*/*z* 311.2345, calculated for C20H32ONa) was a hydroxyl classified as cembratetriaene-ol. Additionally, the stable ion of compound **T6** (*m*/*z* 343.2238, calculated for C_20_H_32_O_3_Na) had oxygen, which indicated that there might be an epoxy group. Thus, **T6** was speculated to be cembratriene-diol oxide.

In this study, 11 compounds (**T1**–**T11**) were identified from the glandular trichome exudates of the air/sun-cured tobacco germplasms. A comparison of data on plant terpenoids reported in the literature [26,33] revealed that the compound **T4** was first reported for natural tobacco.

#### 2.1.2. GC-MS Assay

Using sucrose octa-acetate as an internal standard, sucrose esters in the extracts were semi-quantitatively detected by GC-MS in SIM mode. According to the fragmentation characteristics of sucrose esters in EI mass spectrometry, the characteristic fragment ions of SE(I), SE(II), SE(III), SE(IV), and SE(V) were *m*/*z* 443, 457, 471, 485, and 499, respectively. The difference between the molecular weight of each homolog was 14. As shown in Figure 2, the retention time, shape, and the number of different peaks were highly similar for the different germplasms. In this chromatogram, several chromatographic peaks appeared during the retention time (between 22 and 32 min). Accordingly, the characteristic ions were compared with those in a previous report [34], and the types of sucrose esters corresponding to each peak (SE I–SE V) were determined. The results of the SE contents of 80 air/sun-cured tobacco germplasms are presented in the attached Table A2.

### 2.2. Identification of Antifungal Activity in the Tobacco Glandular Trichome Extracts

#### 2.2.1. Activity Assay of Glandular Trichome Extracts from Four Different Parts of the Tobacco Plant

Four treatments were evaluated in this experiment, namely, glandular trichome extracts from the upper, middle, and lower leaves, and flowers of Zhijinheidiaoba. The components and contents of the glandular trichome extracts from the different parts were determined by UPLC. As shown in Figure 3a, the glandular trichome exudates from different parts of the tobacco leaves and flowers presented the same compositions but in different ratios. The component content of the flowers was 5–8 times higher than that of the middle/lower leaves of the tobacco plant. The inhibitory activities of the different fresh weight concentrations against *B. cinerea* were measured using the mycelial growth rate method, and the results showed that the inhibitory rates of the four treatments increased with the extract concentration. At similar fresh weight concentrations, the inhibitory rates of the glandular trichome extracts from the upper leaves and flowers against *B. cinerea* were significantly higher than those from the middle and lower leaves. At low concentrations (fresh weight ≤ 10 mg/mL), the inhibitory rate of the lower leaf trichome extracts was the lowest, and significant differences were not observed between the middle and lower leaf trichome extracts. However, at higher concentrations (>10 mg/mL), significant differences were observed in the inhibitory rate between the middle and lower leaf trichome extracts (Figure 3b). Virulence analyses of the extracts with different fresh weight concentrations on the growth of *B. cinerea* (Figure 3c) showed that the linear regression equation for the logarithm of extract concentration with different treatments and inhibition rate probability was well fit. Of note, the EC50 values of the extracts from the upper, middle, and lower leaves and flowers were 6.15, 9.57, 14.53, and 3.59 mg/mL, respectively. These differences were highly significant.

#### 2.2.2. Identification of Antifungal Activities among the 80 Glandular Trichome Extracts

According to the results of Section 2.2.1, the glandular trichome extracts at a concentration of 2 mg/mL fresh weight inhibited *B. cinerea*. The antifungal rate of the glandular trichome extracts from the four treatments at the five tested concentrations was over 20.00%. When the fresh weight concentration of the glandular trichome extract ranged from 5 to 20 mg/mL, the antifungal rate against *B. cinerea* changed rapidly. Therefore, the fresh weight concentration was set at 10 mg/mL for the evaluation of antifungal activity by the 80 tobacco germplasm resources. The carbendazim (CBZ) is a positive control (0.06 μg/mL). The results revealed that under the same fresh weight concentration, the inhibitory effects of the 80 glandular trichome extracts against *B. cinerea* were significantly different, and the inhibitory rate ranged from 13.93 to 100.00%, with an average of 64.77% (Figure 4). Among them, 59 germplasm resources (75% ratio of the tested resources) had an antimicrobial activity rate of more than 50.00%, and four germplasm resources (No. X4, X63, X79, and X80; 5% ratio of the tested resources) had an antimicrobial activity rate of more than 95%.

### 2.3. OPLS Model Analysis of Antimicrobial Activity and Secretion Components

The OPLS regression model was established using a dataset of 80 samples (80 × 16). Based on the OPLS model, the R^2^X and Q^2^ values were 0.742 and 0.501, respectively (Table 2). To improve the predictive capacity, sample outliers were identified and excluded using DMOdx. Subsequently, 30 samples (30 × 16) were selected based on R^2^X, Q^2^, and RMSPCV values of 0.861, 0.747, and 0.1336, respectively. At this point, the model (30 × 16) had a stronger explanatory ability for antifungal activity. Afterward, the OPLS method was used to analyze the spectrum-effect relationship between secretion components and antimicrobial activity. The regression coefficient (RC) and variable-introduced projection (VIP) methods were used to construct the OPLS model (R^2^X = 0.86; Q^2^ = 0.747), which showed good fitting and predictive ability when the parameters were above 0.5. A scatter plot of the scores (Figure 5A) displayed the separation ability of the sample according to the antifungal rates. The higher the absolute RC or VIP value, the higher the contribution of antifungal activity. The established OPLS model was used to predict the weights of the antifungal components. There was a positive correlation between the nine peaks and the antifungal rate (Figure 5B), and the correlation coefficients from high to low corresponded to Peak7, Peak8, SE(III), Peak5, Peak9, SE(IV), Peak11, Peak6, and Peak1. A negative correlation was observed between seven peaks and antifungal activity, and the correlation coefficients from high to low corresponded to Peak3, Peak10, Peak4, SE(II), SE(V), SE(I), and Peak2. The higher the VIP value, the more significant the degree of influence (Figure 5C). Among them, Peak6, Peak7, Peak8, Peak9, SE(III), and SE(IV) had significant positive effects on the inhibition rates (VIP ≥ 1). The contributions to antimicrobial activity were in the order Peak7 = Peak8 > SE(III) > Peak9 > SE(IV) > Peak6.

## 3. Discussion

### 3.1. Tobacco Glandular Trichomes

Tobacco glandular trichomes are a type of epidermal trichome, and they account for approximately 85% of the total epidermal trichomes. According to the “with/without secretions” criterion, tobacco glandular trichomes can be divided into glandular and non-glandular trichome types, and glandular types can be further divided into secreting and non-secreting strains [22]. Secretions containing terpenes and carbohydrate esters have antifungal, insecticidal, and antiviral activities [35,36]. The composition and content of secretions are largely influenced by the genotype, environment, organ, and organ location [37]. The results revealed that the composition of glandular trichome secretions from tobacco flowers was the same as that from leaves, although the contents were significantly different. The antifungal activities of ethanolic extracts from the glandular trichomes from the upper leaves and flowers were significantly higher than those from tobacco leaves at the same fresh weight concentration. Tobacco upper leaves and flowers are reproductive organs and thus have higher growth, metabolism, glandular density, and glandular trichome secretions than other parts of the leaf. Therefore, in the tobacco production process, the topped flowers and upper leaves present application value as a raw material for botanical pesticides due to their rich secretions.

### 3.2. Excellent Tobacco Resources Evaluated Based on Antifungal Activity

Currently, active ingredients for botanical fungicides are mainly derived from medicinal plants, such as Asteraceae, Persicum, Euphorbiaceae, and Leguminosae. However, these plants have low biomass yields and high extraction costs, which limit their widespread application as botanical fungicides. Tobacco plants are tall, have high biomass yields, and are rich in secondary metabolites, and they can be planted once and produce multiple harvests. In addition, the active ingredient content of different air/sun-cured tobacco germplasms varied from several to hundreds of times. Therefore, the excellent air/sun-cured tobacco germplasms identified here have important exploration value as resources of botanical pesticides.

OPLS discriminant analysis also showed that α- and β-CBT-diols had the highest contribution to antifungal activity. However, the antifungal rate of Jianpingpiaobayan (X48), which had the highest CBT-diols content, was only 78.88%. This may be related to the X48 germplasm having the highest content of secocembra trihydroxy-diene acid (**T4**), which significantly negatively correlated with antifungal activity. In this study, the OPLS discriminant method was used for modeling and analysis because it can overcome the problem that the ability of the independent variable to explain the dependent variables is easily ignored and avoids the serious consequences of high and multiple correlations among independent variables. Therefore, the results of the study are more reliable than those obtained with a simple correlation analysis. The component analysis clearly showed that the antifungal activity of the extracts was advantageous and effective in the search for new active ingredients. Finally, China has a rich resource of air/sun-cured tobacco resources (>3000) with significant variations in the composition and content of trichome secretions. In particular, Baimaoyan (X80), Huangmaoyan (X79), and CS0708 (X36) selected in this study can provide a material basis for the excavation of anti-fungal ingredients and genetic research on active components.

### 3.3. Mining and Utilization of Antimicrobial Substances

The exploration of new antimicrobial components from plants is a well-established method for the discovery and exploration of new pesticides. To date, research on the antifungal activity of tobacco glandular trichome secretions has focused mainly on α- and β-CBT-diols. Studies have reported that CBT-diols not only have good antifungal activity against apple rot, cucumber anthracnose, gray mold, and other fungal diseases [27,35,38] but also have inhibitory effects on *S. aureus*, *B. subtilis*, *P. vulgaris*, and other bacteria [39]. Moreover, CBT-diols even have a certain control effect on TMV [29,30,40]. The present study revealed that the inhibitory effect on *B. cinerea* differed between the epimers of **T1**, including **T2**, **T3**, **T5**, **T6**, and **T9**; thus, they have application value in the development of antifungal agents. Additionally, sucrose esters play an important role as surfactants in food, medicine, cosmetics, and other fields. Studies on their biological activities have mainly focused on their insecticide and bacterial inhibition abilities [41,42], while their antifungal activity has not been reported. Association analysis showed that sucrose esters SE(III) and SE(IV) had a significant inhibitory effect on *B. cinerea*, while SE(V) showed a significant negative correlation with the inhibitory rate of *B. cinerea*. These terpenoid compounds and sucrose esters with antifungal activity need to be further isolated, purified, and verified.

## 4. Conclusions

This study systematically evaluated the antifungal activity of ethanol extracts of tobacco glandular trichomes against *Botrytis cinerea*. The results showed that glandular trichome secretions from different parts of the tobacco plant exhibited antifungal activity. The extracts from tobacco flowers had higher antifungal activity than those from tobacco leaves at the same fresh weight concentration. In addition, the antifungal activity of glandular trichome extracts varied among different tobacco germplasm, with inhibition rates ranging from 13.93% to 100.00%. Four tobacco germplasms had inhibition rates greater than 95%, including two introduced resources and two local germplasms. The local germplasms, Huangmaoyan and Baimaoyan, had inhibition rates of 99.42% and 100%, respectively, significantly higher than the two introduced germplasms, Cigar No. 5 and Ha20. Finally, the correlation analysis indicated that α-CBT-diol and β-CBT-diol were the major contributors to the antifungal activity, followed by SE(III) and cembratriene-diol oxide. These findings could provide a material basis and technical support for the development of plant-derived fungicides and the efficient utilization of superior tobacco resources.

## 5. Materials and Methods

### 5.1. Materials and Reagents

*B. cinerea* B05.10 fungus was provided by Qingdao Agricultural University (Qingdao, Shandong, China). The 81 air/sun-cured tobacco germplasms were obtained from the National Infrastructure for Crop Germplasm Resources (Tobacco; Qingdao, China) of the Chinese Academy of Agricultural Sciences. The leaves and flowers of the Zhijinheidiaoba germplasm were used to extract glandular trichome secretions. The 80 air/sun-cured tobacco germplasm flowers were used to extract glandular trichome secretions. The names of the 80 air/sun-cured tobacco germplasm are presented in Table 3.

Standards: Standards of α-CBT (98% purities, isolated from tobacco by our laboratory) [27], β-CBT (98% purities, isolated from tobacco by our laboratory) [27], and CBT-ol (98% purities, isolated from tobacco by our laboratory) [27] were isolated from tobacco by our laboratory. The cis-abienol standard (96% purity, Beijing Bailingwei Technology Co., Ltd., Beijing, China) was purchased from Shanghai Zhenzhun Biotech Co., Ltd. (Shanghai, China), the sucrose octa-acetate standard (purity: 98%) was purchased from Sigma-Aldrich Trading Co., Ltd. (Shanghai, China), and the carbendazim (CBZ) was purchased from Jiangsu Lanfeng Biochemical Co., Ltd. (Xuzhou, China).

Reagents: Acetonitrile was chromatographically pure and purchased from Merck & Co., Inc. (Kenilworth, NJ, USA). Analytical-grade ethyl acetate, NN-dimethylformamide, 95% ethanol, and anhydrous sodium sulfate were purchased from Sinopharm Chemical Reagent Corporation, (Shanghai, China). Bis (trimethylsilyl) trifluoroacetamide (purity: 98%) was purchased from Acros Organics (Carlsbad, CA, USA). Ultrapure water was also used.

### 5.2. Methods

#### 5.2.1. Planting of Germplasm Resources

The Zhijinheidiaoba germplasm was planted at the Luozhuang Experimental Station in Shandong Province, and 80 air/sun-cured germplasms were planted at the Cigar Scientific Research and Testing Station in Sichuan Province in 2021. The experimental field is flat and has uniform soil fertility. Plants were cultivated in a row spacing of 110 cm and a plant spacing of 45 cm. Two rows of each replicate were planted, with 20–25 plants per row and three replicates. Field management was carried out according to local practices.

#### 5.2.2. Extraction of Glandular Trichome Secretions

The glandular trichome secretions were extracted from the upper (14–15 leaves), middle (10–11 leaves), and lower (4–5 leaves) leaves and flowers of Zhijinheidiaoba at the full-bloom stage. Glandular trichome secretions were also extracted from the flowers of the 80 air/sun-cured tobacco germplasms at the full-bloom stage. Five plants were included in each replicate, and three independent replicates were performed. Twenty flowers from five inflorescences were obtained during the growth period as shown in the rectangular box in Figure 6, and they were weighed, placed in 200 mL conical flasks containing 100 mL 95% ethanol, and then shaken for 60 s [27]. The resulting extracts were stored at 0–4 °C until further analysis.

#### 5.2.3. Detection Assays

Ultra-performance liquid chromatography quadrupole time of flight-mass spectrometer (UPLC-Q-TOF MS) (Bruker, Billerica, MA, USA)) analysis: Chromatographic separation was performed on an ACQUITY UPLC BEH C18 column (2.1 mm × 100 mm; 1.7 μm particle size) with a UV-Vis detector. The mobile phase consisted of acetonitrile (solvent A) and ultrapure water (solvent B), and the gradient program was 0–6 min, 20–80% A; 6–11 min, 100% A; and 11–16 min, 80–20% at a flow rate of 0.3 mL/min, and column temperature of 35 °C. For mass detection, an electrospray ionization (ESI) source was operated in positive mode with a scan range of 50–1000 *m*/*z*.

GC-MS analysis: The detection conditions were as follows: helium carrier gas in an HP-MS column (30 m × 250 μm × 0.25 μm) at a flow rate of 1.0 mL/min, inlet temperature of 290 °C, split stream with a 3:1 split ratio, and injection volume of 1 μL. The initial column temperature of 230 °C was increased to 270 °C (rate: 5 °C/min) and held for 2 min. Subsequently, the temperature was increased to 290 °C (rate: 1 °C/min) and held for 4 min. The mass spectrum transmission line temperature was 280 °C; the four-stage rod temperature was 150 °C; EI source temperature was 230 °C; electron energy was 70 eV; and mass scanning range was 30–550 amu.

Quantitative analysis: Contents of α-CBT, β-CBT, CBT-ol, and cis-abienol were calculated by corresponding standard curves. The contents of the other gradients detected by UPLC-Q-TOF MS were calculated using the formula Cn = Sn/S β-CBT-diol × Cβ-CBT-diol, where Cn is the content of peaks without standards; Cβ-CBT-diol is the content of β-CBT-diol; Sn indicates the peak area of peaks without standards; and Sβ-CBT-diol indicates the peak area of β-CBT-diol. Sucrose octa-acetate was used as an internal standard for the semi-quantitative analysis of sucrose esters.

#### 5.2.4. Antifungal Activity

The secretions of the glandular trichomes extracts from four different parts (lower, middle, and upper leaves and flowers) of Zhijinheidiaoba were calculated on a fresh weight basis and then blown dry with nitrogen. The four samples were dissolved in 95% ethanol to obtain the same fresh weight concentrations and then diluted with 95% ethanol to concentrations of 20, 16, 12, 8, 4, and 2 mg/mL fresh weight. The corresponding volumes of the 80 extracts were calculated at a concentration of 1.00 g/mL fresh weight and subsequently blown dry with nitrogen. Each extract was dissolved in 1 mL of 95% ethanol, and the same volume of 95% ethanol was used as a blank control. The extracts were then aseptically added to 99 mL of potato glucose agar (PDA) medium (4.6%) at 45 °C, and the final fresh weight concentration was 10.00 mg/mL. Carbendazim (CBZ) was utilized as a positive control at a concentration of 0.06 μg/mL. The PDA medium was poured into an average of four Petri dishes. *B. cinerea* (6 mm in diameter) was placed in the center of the Petri dish, incubated at 25 °C, and cultured for three days. The maximum diameter of the control colony was measured when it reached 75–85% of the plate using the crisscross method. The inhibition rates of the different extracts were calculated according to the percentage change in colony diameter compared to the control group. The antifungal rate was calculated as [(mean colony diameter of control − mean colony diameter of treatment)/(mean colony diameter of control − 6)] × 100%.

### 5.3. Statistical Analysis

Data entry and test significance were performed in Excel (version 2019) and SPSS software (version 23.0), respectively. OPLS multivariate data analysis (SIMCA-P, version 14.0) was used to construct a regression model of trichome exudates and determine the in vitro antifungal activity of the extracts.

## Figures and Tables

**Figure 1 plants-13-01997-f001:**
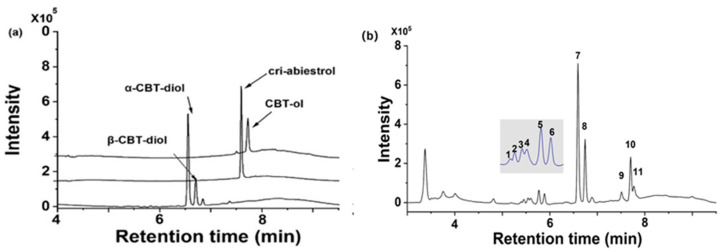
UPLC chromatograms of the diterpenoid components from the glandular trichome extracts of the sun/air-cured tobacco. (**a**) UPLC chromatogram of the standards α-CBT, β-CBT, cri-abiestrol, and CBT-ol. (**b**) UPLC chromatograms of glandular trichome secretions from tobacco.

**Figure 2 plants-13-01997-f002:**
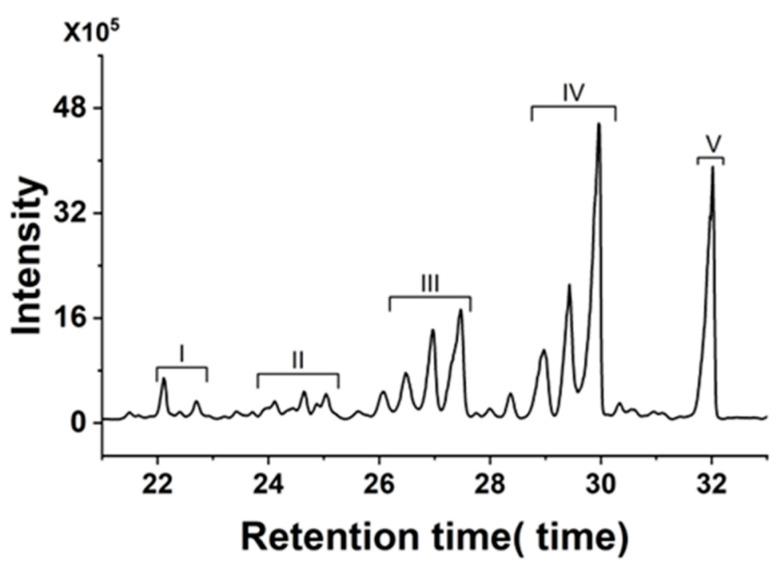
GC chromatograms of sucrose esters from glandular trichome extracts of sun/air-cured tobacco.

**Figure 3 plants-13-01997-f003:**
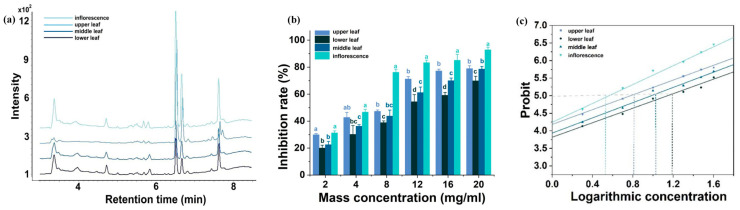
UPLC chromatograms of the extracts from different parts of the tobacco plant and inhibitory effects on mycelial growth of *Botrytis cinerea*. The lightest blue, second lightest blue, royal blue, and navy colors indicate the flowers, upper leaf, middle leaf, and lower leaf, respectively. (**a**) UPLC stacking diagram of glandular trichome extracts from different parts of the tobacco plant. (**b**) Antifungal activity of different treatments against *B. cinerea* at different concentrations. (**c**) Relationship between the antifungal probability and logarithmic concentration.

**Figure 4 plants-13-01997-f004:**
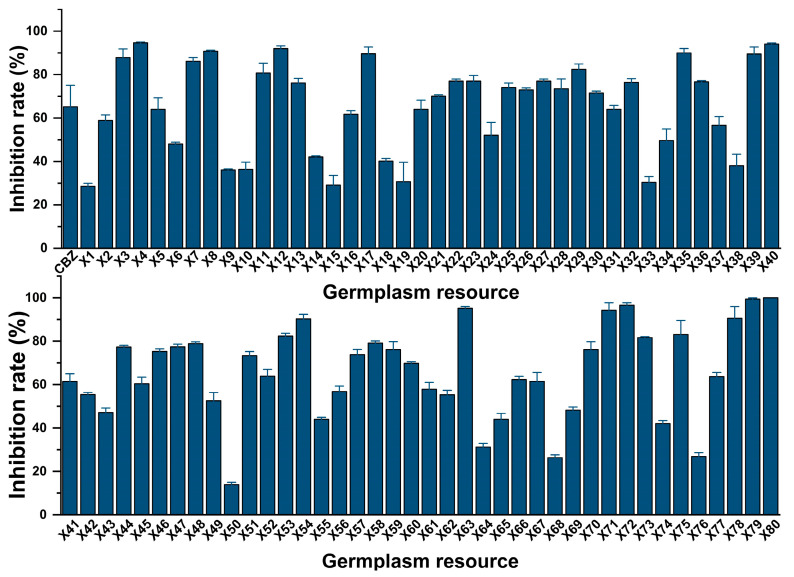
Inhibitory activity against *B. cinerea* of the glandular trichome extracts from flowers of 80 tobacco germplasm resources. CBZ: carbendazim.

**Figure 5 plants-13-01997-f005:**
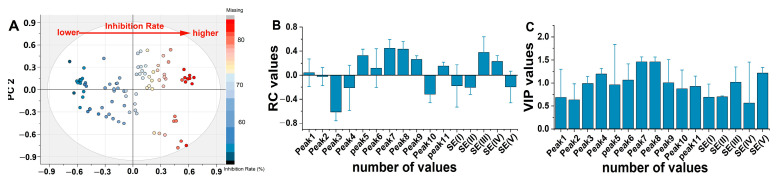
OPLS model analysis of the glandular trichome extract fractions from flowers and antifungal activity. (**A**) Scores plot; (**B**) RC values; and (**C**) VIP values.

**Figure 6 plants-13-01997-f006:**
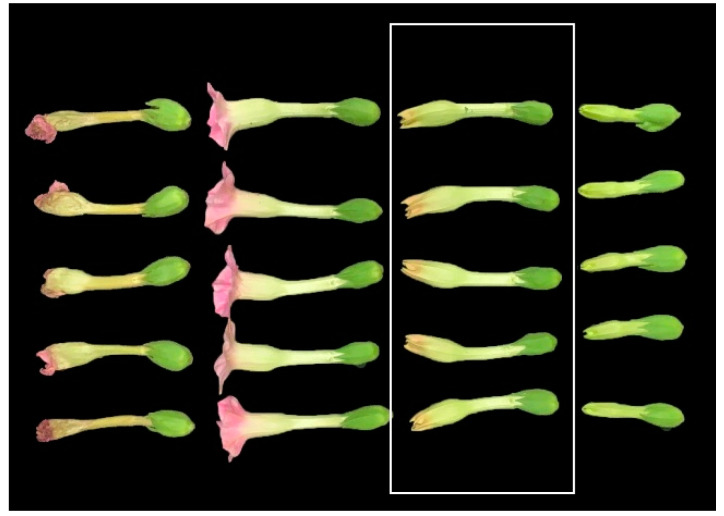
Tobacco flowers at different stages of development.

**Table 1 plants-13-01997-t001:** Retention times, formulae, mass errors, and fragment ions of components from different peaks by UPLC-Q-TOF-MS.

Peak	Formula	Compound	tR (min)	Error (ppm)	Score	*m*/*z*
Peak1	C_20_H_34_O_4_	**T1**	4.3 min	−1.4	100	361.2341, 303.2304, 287.2356
Peak2	C_20_H_34_O_4_	**T2**	4.4 min	−1.6	100	361.2340, 303.2310, 287.2366
Peak3	C_20_H_34_O_4_	**T3**	4.5 min	1.8	84.46	361.2396, 303.2420, 287.2361
Peak4	C_22_H_40_O_5_	**T4**	4.7 min	−1.9	89.67	407.2768, 349.2734, 331.2226, 286.2242
Peak5	C_20_H_34_O_3_	**T5**	4.9 min	−2.4	89.67	345.2492, 305.2476, 287.2361, 269.2351
Peak6	C_20_H_32_O_3_	**T6**	5.1 min	1.8	100	343.2238, 303.2311, 285.2209
Peak7	C_20_H_34_O_2_	**T7**	6.6 min	0.6	100	329.2453, 289.2523, 271.2417
Peak8	C_20_H_34_O_2_	**T8**	6.9 min	−0.2	100	329.2451, 289.2522, 271.2417
Peak9	C_20_H_32_O	**T9**	8.4 min	1.2	100	311.2345, 271.2414
Peak10	C_20_H_34_O	**T10**	8.6 min	1.4	100	313.2446, 273.2521, 255.241
Peak11	C_20_H_34_O	**T11**	8.8 min	1.3	100	313.2502, 273.2537

**Table 2 plants-13-01997-t002:** Comparison of orthogonal projections with latent structure models before and after variable selection.

Matrix	nLV	R^2^X	Q^2^	RMSECV
80 × 16	4	0.742	0.501	0.1892
60 × 16	4	0.762	0.593	0.1796
40 × 16	3	0.775	0.639	0.1877
30 × 16	2	0.861	0.747	0.1336
20 × 16	2	0.881	0.643	0.1511

**Table 3 plants-13-01997-t003:** Eighty air/sun-cured tobacco germplasm resources.

No.	Name	Type	Source	No.	Name	Type	Source
X1	Havana-1	cigar	introduced	X41	Tangfeng	sun-cured	local
X2	Havana 1	cigar	introduced	X42	Jiangyouyan	sun-cured	local
X3	Beinhart 1000-1	cigar	introduced	X43	Wushanxiaolanyan	sun-cured	local
X4	Criollo Salteno 11	cigar	introduced	X44	Tiebanqing	sun-cured	local
X5	S-2	cigar	introduced	X45	Meitanshaiyan	sun-cured	local
X6	Havana IIc	cigar	introduced	X46	Hefengheiyan	sun-cured	local
X7	Zrenjanin	cigar	introduced	X47	Liufengmaobayan	sun-cured	local
X8	Yinnixuejiabaopi	cigar	introduced	X48	Jianpingpiaobayan	sun-cured	local
X9	112–117	cigar	introduced	X49	Dayeziqingyan	sun-cured	local
X10	Bad Geudertheimer Landsorte	cigar	introduced	X50	Lichuanmaoyan	sun-cured	local
X11	Begej	cigar	introduced	X51	Zhushandaliuzi	sun-cured	local
X12	Connecticut Broad Leaf	cigar	introduced	X52	Fengjiedamaoyan	sun-cured	local
X13	Connecticut Shade	cigar	introduced	X53	Shai9118	sun-cured	breeding
X14	E 18	cigar	introduced	X54	Shifangpipaliu	sun-cured	local
X15	Geudetthelmex	cigar	introduced	X55	Shiyan1	sun-cured	local
X16	Hanica	cigar	introduced	X56	Kuiliu	sun-cured	local
X17	Havana 211	cigar	introduced	X57	Bamaoliu	sun-cured	local
X18	Havana 510	cigar	introduced	X58	Mianzhushaiyan	sun-cured	local
X19	Manila	cigar	introduced	X59	Bashan1	sun-cured	local
X20	Havana	cigar	introduced	X60	Chongzhoushaiyan2	sun-cured	local
X21	Tuerqixueji	cigar	introduced	X61	Quanyan	sun-cured	local
X22	Conn Shade	cigar	introduced	X62	Zhouyan	sun-cured	local
X23	Dexue 1	cigar	introduced	X63	Xuejia5	sun-cured	introduced
X24	Dexue2	cigar	introduced	X64	New Havana IIc	cigar	introduced
X25	Dexue 3	cigar	introduced	X65	Comstock Spanish	cigar	introduced
X26	Habana92	cigar	introduced	X66	Mont Calme Brun	cigar	introduced
X27	Cubra-Brazil	cigar	introduced	X67	Trapesond 288	cigar	introduced
X28	Kangzhoukuoye	cigar	introduced	X68	CA0705	cigar	breeding
X29	OLOR	cigar	introduced	X69	CA0709	cigar	breeding
X30	Duominijiachangxin	cigar	introduced	X70	Nilajiaguachangxin	cigar	introduced
X31	Duominijiaduanxin	cigar	introduced	X71	Besuki	cigar	introduced
X32	MFPP	cigar	introduced	X72	Ha20	cigar	introduced
X33	MFZS	cigar	introduced	X73	Criollo	cigar	introduced
X34	CP2011	cigar	introduced	X74	Ha12	cigar	introduced
X35	MSCA	cigar	introduced	X75	Ha19	cigar	introduced
X36	CS0708	cigar	breeding	X76	Jilindabaihua	sun-cured	local
X37	Shangzhiyiduohua	air-cured	local	X77	Mulengdaqingjin	sun-cured	local
X38	Shandongdaye	sun-cured	local	X78	Juanyeshaiyan	aromatic	local
X39	Xinbinxiaotuanye	sun-cured	local	X79	Huangmaoyan	sun-cured	local
X40	Liaoduoye	sun-cured	local	X80	Baimaoyan	sun-cured	local

## Data Availability

Data are contained within the article.

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
