# Peer review of "Application and Evaluation of the Antifungal Activities of Glandular Trichome Secretions from Air/Sun-Cured Tobacco Germplasms against Botrytis cinerea"

_plants, 2024, doi:10.3390/plants13141997_

Round 1

Reviewer 1 Report

Comments and Suggestions for Authors

I think the paper is interesting, and that the authors carried out a great effort in investigating such a high number of samples.

However, I think some point need to be clarified:

The authors reported the different antifungal activity of each extract from the flowers of 80 tobacco germplasm resources. The same think has not been done for the phytochemical characterization. It is not clear to which sample refer the analyses reported in figures 1 and 2, as well as table 1.

Consistently with the different biological properties exerted by different germoplasms, some differences could be  identified among the phytochemical composition of the different extracts.

Looking at the statistical analyses the authors carried out the authors probably performed such analyses but they are not reported in the paper. The phytochemical differences should be highlighted as well as the differences in the biological potential.

Moreover, in the introduction section, the authors should better discuss previous studies focusing on the tobacco resistance against Bothytis cinerea.

Author Response

Comment 1: The authors reported the different antifungal activity of each extract from the flowers of 80 tobacco germplasm resources. The same think has not been done for the phytochemical characterization. It is not clear to which sample refer the analyses reported in figures 1 and 2, as well as table 1.

Answer 1: Thank you for your feedback.  Figure 1(a) is the chromatogram of the standard. Figures 1(b) and Figure 2 are schematic diagrams, showing that up to 11 components (UPLC) and 5 components (GC-MS) can be detected, respectively. Table 1 lists the 11 components detected in Figure 1b, labeled from T1 to T11.

Comment 2: Consistently with the different biological properties exerted by different germplasms, some differences could be identified among the phytochemical composition of the different extracts.

Answer 2: Thank you for your feedback. We have supplemented 2 Tables: The contents of the identified components of the 80 air/sun-cured tobacco germplasms (Table S1, Table S2), These two tables illustrate the variations in the component content among 80 different germplasm resources.

Comment 3: Looking at the statistical analyses the authors carried out the authors probably performed such analyses but they are not reported in the paper. The phytochemical differences should be highlighted as well as the differences in the biological potential.

Answer 3: Thank you for your feedback. We have added supplementary Tables S1 and S2 about detecting 16 component contents of 80 germplasm resources. The tables highlight the variations in content among different components and different germplasms.

Comment4: Moreover, in the introduction section, the authors should better discuss previous studies focusing on the tobacco resistance against Bothytis cinerea.

Answer 4: Thank you for your feedback. Our study focuses on evaluating the application of tobacco glandular trichome secretions in combating fungal diseases, rather than investigating the resistance genes of tobacco against Botrytis cinerea. While we acknowledge the importance of studies on tobacco resistance against Botrytis cinerea, our primary objective is to explore the antifungal activity of glandular trichome secretions.

Reviewer 2 Report

Comments and Suggestions for Authors

The paper is interesting. However several corrections are required.

Major concerns

1)   A major concern in this manuscript is the lack of a standard positive drug (standard antifungal drug, fungicide) in the assays. For example in Table 4, no results of a standard positive drug were included. This is mandatory for any pharmacological assay.

You can read: Raorane, C.J., Raj, V., Lee, J.H., Lee, J. (2022). Antifungal activities of fluoroindoles against the postharvest pathogen Botrytis cinerea: In vitro and in silico approaches. Int. J. Food Microbiol. 62, 109492.

You could use carbendazim or any other.

2)   Lines 12-13: change “The glandular trichomes of tobacco leaves and flowers can secrete abundant compounds, such as cebradanes, labdanes, and saccharide esters. These secondary metabolites show high biological activity, including  antimicrobial, insecticidal, and antioxidant  activity.”

Note of this Reviewer: Here there is a conceptual error, because the glandular trichomas do not secrete ‘secondary metabolites’ on their own. They produce “a secretion” and this secretion contains different secondary metabolites.

So, change the paragraph to:

“The secretions of the glandular trichomes of tobacco leaves and flowers contain abundant secondary metabolites of different type, such as cebradanes, labdanes, and saccharide esters. They have shown interesting biological properties, such as antimicrobial, insecticidal and antioxidant activities”.

Line 256: You say “Secretions of terpenes and carbohydrate esters”. Change to “Secretions containing terpenes and carbohydrate esters”

 3)   Line 59: change “Tobacco (genus, Nicotiana; family, Solanaceae)” to “Tobacco (Nicotiana tabacum L. (Solanaceae)”

 4)   Line 60: and has the highest biological yield in nature?? What do you mean?. Please re-phrase

 5)   Lines 64-65: again there is a conceptual error here. Change “that represent the most abundant glandular trichome secretions of tobacco germplasms” to “that are the most abundant secondary metabolites in the glandular trichome secretions of tobacco germplasms”

 6)   Line 79: What do you mean with “were ‘systematically’ evaluated.”?. Eliminate the word “systematically”. The same in line 86.

7)   Lines 96-107: The whole paragraph are results. This paragraph must be eliminated from the Introduction and must be shift to the Results section.

 8)   Lines 111, 112: change to “were qualitative and quantitatively analyzed. The qualitative analysis was performed with UPLC-Q-TOF-MS in the ESI-positive mode”. Line 117: change to “Regarding the quantitative analysis, the contents of the components were calculated by using corresponding standard curves.”

9)   Each Table must work on its own. So, in Table 1, you must give the names of the compounds and not put T1, T2 etc.

 10)    Line 315: Please provide the voucher number of the B. cynerea strain. Surely, you deposited it in a Fungi Culture collection.

Minor remarks

Write “activities” (in the title), “spectrometry” (in line 18) “activity” (In line 19), “identify” (in line 29), “regulations” (in line 46), prevent (line 47), “metabolites” (line 52), “characterized” (line 52) in a sole word.

I cannot understand why so many words are cut with a dash. The authors must type the text without clicking the "enter" key. Revise the whole text.

Write “Botrytis cinerea” (in the title, line 19 and throughout the text) in italics. All binomial names must be written in italics. Correct in lines 67, 68 and throughout the text

Lines 22, 23: change “in the antifungal activities of ethanol extracts of tobacco glandular trichomes.” to “among the antifungal activities of the tested ethanol extracts of tobacco glandular trichomes”

Lines 27, 28: Each abbreviation must be clarified in its first mentioning. So, clarify “α-CBT-diol”, “β-CBT-diol” and “SE” in the Abstract and, independently, in the main text

Line 40, add a space: “transmission [4]”. The same in line 64

Line 41: “per” in italics

Line 45: After “Committee” a reference is missing

Line 44, add quotation marks: ‘highly resistant’

Line 51: change “Due to the impacts caused by chemical pesticides” to “Due to this negative impact”

Line 175: “Four treatments” or “Four samples”? Please correct

Line 257: change “biological activity” to “activities”

Line 263: eliminate “This may be due to the content of secretions in the extracts.” This statement is not correct

Line 379. You say “The glandular trichome extracts”. Do you mean: “The trichome secretion extracts”? Please correct

Comments on the Quality of English Language

My corrections are included in the comments on the manuscript

Author Response

comments 1: 1)   A major concern in this manuscript is the lack of a standard positive drug (standard antifungal drug, fungicide) in the assays. For example in Table 4, no results of a standard positive drug were included. This is mandatory for any pharmacological assay.

You can read: Raorane, C.J., Raj, V., Lee, J.H., Lee, J. (2022). Antifungal activities of fluoroindoles against the postharvest pathogen Botrytis cinerea: In vitro and in silico approaches. Int. J. Food Microbiol. 62, 109492.

You could use carbendazim or any other.

Answer 1: This study primarily explores the antifungal activity of components derived from tobacco glandular trichome secretions. The main is to investigate different germplasm resources to develop superior germplasm. By comparing the antifungal activities among various resources, we aim to identify and select superior germplasm.

comments 2:  2)   Lines 12-13: change “The glandular trichomes of tobacco leaves and flowers can secrete abundant compounds, such as cebradanes, labdanes, and saccharide esters. These secondary metabolites show high biological activity, including antimicrobial, insecticidal, and antioxidant activity.”Note of this Reviewer: Here there is a conceptual error, because the glandular trichomas do not secrete ‘secondary metabolites’ on their own. They produce “a secretion” and this secretion contains different secondary metabolites. So, change the paragraph to:“The secretions of the glandular trichomes of tobacco leaves and flowers contain abundant secondary metabolites of different type, such as cebradanes, labdanes, and saccharide esters. They have shown interesting biological properties, such as antimicrobial, insecticidal and antioxidant activities”. Line 256: You say “Secretions of terpenes and carbohydrate esters”. Change to “Secretions containing terpenes and carbohydrate esters”

Answer 2 : Thank you for your comments, I have corrected it.

 comments 3: 3) Line 59: change “Tobacco (genus, Nicotiana; family, Solanaceae)” to “Tobacco (Nicotiana tabacum L. (Solanaceae)”

Answer 3: Thank you for your comments, I have corrected it.

comments 4:  4)   Line 60: and has the highest biological yield in nature?? What do you mean?. Please re-phrase

Answer 4: we revised this sentence that tobacco has a high biomass yield((up to 100 t of leaf biomass per hectare))

comments 5:  5)   Lines 64-65: again there is a conceptual error here. Change “that represent the most abundant glandular trichome secretions of tobacco germplasms” to “that are the most abundant secondary metabolites in the glandular trichome secretions of tobacco germplasms”

Answer 5: We corrected it.

comments 6:  6)   Line 79: What do you mean with “were ‘systematically’ evaluated.”?. Eliminate the word “systematically”. The same in line 86.

Answer 6: I deleted it.

comments 7: 7)   Lines 96-107: The whole paragraph are results. This paragraph must be eliminated from the Introduction and must be shift to the Results section.

Answer 7: the whole paragraph is below the 2 Results

comments 8: 8)   Lines 111, 112: change to “were qualitative and quantitatively analyzed. The qualitative analysis was performed with UPLC-Q-TOF-MS in the ESI-positive mode”. Line 117: change to “Regarding the quantitative analysis, the contents of the components were calculated by using corresponding standard curves.”

Answer 8: Thank you for your advice, we revised that.

comments 9: 9)   Each Table must work on its own. So, in Table 1, you must give the names of the compounds and not put T1, T2 etc.

Answer 9: Thank you for the feedback, T1, T2, etc are the names of the compounds, the detailed characterization of the compounds has been shown in the text.

 comments 10: 10)    Line 315: Please provide the voucher number of the B. cynerea strain. Surely, you deposited it in a Fungi Culture collection.

Answer 10: I added the voucher number of the B.cinerea.

Minor remarks

comments 11: Write “activities” (in the title), “spectrometry” (in line 18) “activity” (In line 19), “identify” (in line 29), “regulations” (in line 46), prevent (line 47), “metabolites” (line 52), “characterized” (line 52) in a sole word.

Answer 11: Thank you for the feedback, I corrected that.

comments 12: I cannot understand why so many words are cut with a dash. The authors must type the text without clicking the "enter" key. Revise the whole text.

Answer 12: Thank you for the feedback, I corrected that.

comments 13: Write “Botrytis cinerea” (in the title, line 19 and throughout the text) in italics. All binomial names must be written in italics. Correct in lines 67, 68 and throughout the text

Answer 13: Thank you for the feedback, I corrected all the Botrytis cinerea to the italics in the text.

comments 14: Lines 22, 23: change “in the antifungal activities of ethanol extracts of tobacco glandular trichomes.” to “among the antifungal activities of the tested ethanol extracts of tobacco glandular trichomes”

Answer 14: Thank you for the feedback, I corrected that.

comments 15: Lines 27, 28: Each abbreviation must be clarified in its first mentioning. So, clarify “α-CBT-diol”, “β-CBT-diol” and “SE” in the Abstract and, independently, in the main text

 Answer 15: Thank you for the feedback, I clarify that in the Abstract.

comments 16: Line 40, add a space: “transmission [4]”. The same in line 64

Answer 16: Thank you for the feedback, I corrected that.

comments 17: Line 41: “per” in italics

Answer 17: Thank you for the feedback, I corrected that.

comments 18: Line 45: After “Committee” a reference is missing

Answer 18: Thank you for the feedback, I added a reference for this information.

comments 19:Line 44, add quotation marks: ‘highly resistant’

Answer 19: Thank you for the feedback, I added it.

comments 20: Line 51: change “Due to the impacts caused by chemical pesticides” to “Due to this negative impact”

Answer 20: Thank you for the feedback, I corrected that.

comments 21: Line 175: “Four treatments” or “Four samples”? Please correct

Answer 21: Thank you for your question. In our study, we are referring to 'four samples' as there are four individual specimens that we analyzed. I have revised it.

comments 22: Line 257: change “biological activity” to “activities”

Answer 22: Thank you for the feedback, I corrected that.

comments 23: Line 263: eliminate “This may be due to the content of secretions in the extracts.” This statement is not correct

Answer 23: Thank you for the feedback, I deleted it.

comments 24: Line 379. You say “The glandular trichome extracts”. Do you mean: “The trichome secretion extracts”? Please correct

Answer 24: I changed it to “the secretions of the glandular trichomes extracts”

Reviewer 3 Report

Comments and Suggestions for Authors

The authors evaluated the antifungal activities of glandular trichome secretions from air/sun-cured tobacco germplasms against Botrytis cinerea using mass spectrometry and antifungal assays. The manuscript is well-written and can be published after minor revision. Please find my specific comments below:

Methods section:

Some standards were isolated in your laboratory, please add some data about how those standards were isolated and confirmed.

Extraction method: how did you choose extraction parameters and explain solvent choice? Is 60 long extraction sufficient?

Please add short conclusion with main findings, limitation and some suggestions for further studies

Author Response

The authors evaluated the antifungal activities of glandular trichome secretions from air/sun-cured tobacco germplasms against Botrytis cinerea using mass spectrometry and antifungal assays. The manuscript is well-written and can be published after minor revision. Please find my specific comments below:

Methods section:

comments 1: Some standards were isolated in your laboratory, please add some data about how those standards were isolated and confirmed.

Answer 1: Thank you for your comments, I added more information about standards in 4.1 Materials and Reagents, and also added references about how isolated and purified standards are in our laboratory.

comments 2: Extraction method: how did you choose extraction parameters and explain solvent choice? Is 60 long extraction sufficient?

Answer 2: Thank you for your comments, We have done a lot of research on extraction methods of glandular trichome secretions in our laboratory and already published papers. I have added related references. Especially with an extraction time of 60 seconds, not only does it fully extract glandular trichome secretions, but it also prevents the extraction of tobacco components.

Comment 3: Please add short conclusion with main findings, limitation and some suggestions for further studies

Answer 3: Thank you for your comments, We added a short conclusion in 5 conclusions.

Round 2

Reviewer 1 Report

Comments and Suggestions for Authors

The authors addressed the comments and suggestions.

Author Response

Comment1: 

Yes Can be improved Must be improved Not applicable
Does the introduction provide sufficient background and include all relevant references? ( ) (x) ( ) ( )

Answer1: Thank you for your comments, We have improved the relevant introduction section.

Reviewer 2 Report

Comments and Suggestions for Authors

The authors did not add a standard positive drug and thus, the work diminishes its quality.

Below you will find the requirement of this Reviewer in the original version and the evasive response of the authors. 

Comments 1: 1)   A major concern in this manuscript is the lack of a standard positive drug (standard antifungal drug, fungicide) in the assays. For example in Table 4, no results of a standard positive drug were included. This is mandatory for any pharmacological assay.

You can read: Raorane, C.J., Raj, V., Lee, J.H., Lee, J. (2022). Antifungal activities of fluoroindoles against the postharvest pathogen Botrytis cinerea: In vitro and in silico approaches. Int. J. Food Microbiol. 62, 109492.

You could use carbendazim or any other.

Answer 1: This study primarily explores the antifungal activity of components derived from tobacco glandular trichome secretions. The main is to investigate different germplasm resources to develop superior germplasm. By comparing the antifungal activities among various resources, we aim to identify and select superior germplasm.

I recommend to authors to add a standard fungicide 

Author Response

Comments 1: Comments 1: 1)   A major concern in this manuscript is the lack of a standard positive drug (standard antifungal drug, fungicide) in the assays. For example in Table 4, no results of a standard positive drug were included. This is mandatory for any pharmacological assay.

You can read: Raorane, C.J., Raj, V., Lee, J.H., Lee, J. (2022). Antifungal activities of fluoroindoles against the postharvest pathogen Botrytis cinerea: In vitro and in silico approaches. Int. J. Food Microbiol. 62, 109492.

You could use carbendazim or any other.

Answer 1: Thank you for your suggestion. We have added the positive control experiment of Carbendazim and made some additions to the Results (Table 4) and Materials( 5.1 Materials and reagents)

Round 3

Reviewer 2 Report

Comments and Suggestions for Authors

The authors have met my requirement of adding a standard positive drug and now the manuscript is ready for publication